# Examining non-linearity in the association between age and reported opioid use in different socioeconomic strata: cohort study using Health Survey for England waves from 1997 to 2014

Magdalena Nowakowska [ORCID],[1,2,3] Salwa S Zghebi [ORCID],[1,4] Li-Chia Chen,[5] Darren M Ashcroft,[1,3,5] Evangelos Kontopantelis [ORCID] [1,2]

For numbered affiliations see end of article.

**Correspondence to**
Professor Evangelos Kontopantelis;
e.kontopantelis@manchester.ac.uk

## ABSTRACT

**Background** Age and socioeconomic status (SES) predict several health-related outcomes, including prescription opioid use. Contrasting findings from previous literature found higher prevalence of opioid use in both people over 65 years old and the working-age population of 35–55 years old. This study aimed to analyse if the association between age and opioid use is non-linear and differs in adults with different SES levels.

**Methods** This cohort study used the Health Survey for England waves 1997–2014 data to investigate the shape of the correlation between reported opioid use and income decile, employment status and educational level. A semiparametric Generalised Additive Model was employed, so that linearity of correlation was not assumed. The shape of the relationship was assessed using the effective degrees of freedom (EDF).

**Results** Positive correlation between age and reported opioid use, more linear in people in the highest income decile (EDF: 1.01, p<0.001) and higher education (EDF: 2.03, p<0.001) was observed. In people on lower income and with lower levels of education, the highes probability of reported opioid use was at around 40–60 years old and slowly decreased after that. Higher income decile and higher levels of education were predictors of a lower probability of reported opioid use (OR: 0.27, 95% CI: 0.21 to 0.36 and OR: 0.48, 95% CI: 0.41 to 0.57, respectively). There was no statistically significant difference in opioid use between employed and unemployed people.

**Conclusion** The relationship between age and the probability of prescribed opioid use varies greatly across different income and educations strata, highlighting different drivers in opioid prescribing across population groups. More research is needed into exploring patterns in opioid use in older people, particularly from disadvantaged socioeconomic backgrounds.

## INTRODUCTION

Opioid painkillers are effective analgesics that can provide crucial pain relief for patients with acute or cancer pain, which are often still used for the treatment of chronic non-cancer

### STRENGTHS AND LIMITATIONS OF THIS STUDY

⇒ The use of Generalised Additive Models to investigate the relationship between age and reported opioid use within different socioeconomic strata allowed exploration of the correlation without the assumption of linearity.

⇒ Due to limitations in recorded data, the distinction between type, strengths and duration of opioids was not accounted for.

⇒ Survey data used in this study are sensitive to self-reporting bias.

⇒ The low prevalence of opioid users limited the number of covariates used in the statistical model.

pain, although evidence for the latter indicates higher risk of short-term harms and limited or no benefits compared with non-opioid therapy.[1] Extensive research suggests a dose and potency-dependent association between prescribed opioid use and several adverse severe outcomes in patients who persistently use them.[2][3] Prescribed opioid use, as well as the associated adverse effects, varies across individuals and communities.[4][5] A growing body of academic literature focuses on the biological, psychosocial and institutional factors that influence the likelihood of prescribed opioid use. Socioeconomic status (SES), which can be defined through several measures, including income, occupational status and education, was found to be closely associated with opioid prescribing[5] as well as female gender,[6] ethnicities[7] and presence of psychological comorbidities.[8] Age is one of the strongest predictors of health and health outcomes, and the factor most commonly included in predictive models. However, the association between age and prescribed opioid use is not necessarily linear.

Older people are more likely to experience complex health problems and often in need of multiple prescribed medications.[9] Several studies found that older people are more likely to use opioid analgesia.[10] This could be partially explained by the opioids' recommended indications in palliative care and the higher prevalence of painful conditions in older people.[11] In contrast, some studies observed a higher prevalence of prescribed opioid use in 35–49year olds.[12] Working-age adults may be more likely to be exposed to trauma and experience occupation-related pain. In a UK-based study of prescribed opioid use, the highest proportion of new users of weak and moderate opioids were aged 35–54 years old, whereas over 50% of new users of strong opioids were 75 years and older.[13]

The relationship between age and opioid analgesia need is complex and influenced by many factors, including race, underlying mental health conditions and socioeconomic circumstances. A nearly 300% difference in opioid prescription prevalence was observed across the race/ethnicity-income gradient in the USA.[14] In Sweden, the highest risk of prescribed opioid use was observed in medium-income women aged 65 years or older, living alone and with psychological distress; whereas the lowest risk was observed in low-income men aged 18–44, living alone and without psychological distress.[15] SES, including income, employment status and education level, can determine persons living and working conditions, access and interactions with the healthcare system, and overall health status, changing personal health and opioid-related life trajectories. This study hypothesises that the association between age and reported prescribed opioid use varies by SES level. We used Generalised Additive Models (GAMs) to examine this hypothesis, a regression analysis method that allows exploration of the non-linearity of the relationship between age and prescribed opioid use within groups of different income, employment status and education levels. Different associations between age and opioid prescribing, across socioeconomic strata, would potentially highlight the presence of different health needs and drivers for prescribing. This would have implications for research, since we would need to understand the underlying cause or causes of this variation, to inform policy and improve patient care.

## METHODS
### Data source
This cohort study used the Health Survey for England (HSE), a national annual survey designed to monitor the nation's health and care trends.[16] Participants were offered a nurse visit during which several questions about medication use and measurements such as height and weight are recorded. Data on opioid use were available for extraction from surveys conducted between 1997 and 2014, and therefore, these data were analysed in this study. The HSE surveys are reviewed yearly by the Research Ethics Committee (East Midlands—Nottingham 2), and informed consent was obtained from all participants.

All data used in this analysis are freely available and no further approvals are required.[17]

### Study population
Participants in the HSE are randomly chosen, for each wave, from all private households' addresses in England. Participants aged 18 and older with no reported diagnosis of cancer who completed the nurse interview part of the survey were included in the study. At a number of annual waves (1997–2014), a boost sample of a population with specific characteristics was collected. To minimise bias, we excluded the boost sample of people living in care homes which were included in the 2000-year and 2005-year waves. The ethnic minority boost sample included at waves 1999 and 2004 was included in the analysis.

### Definition of opioid use
Participants were asked if they take any prescribed medications and, if yes, permission was sought for a nurse to see its container and to record the name of the medication. Up to 22 prescribed medications were recorded using the British National Formulary (BNF) Codes. Medications coded as opioid analgesics (BNF code 40702) were included in this study. If at least on opioid analgesic was recorded, participant was coded as opioid users, and otherwise, they were coded as not an opioid users.

### SES measures
SES was measured using three commonly used indicators: income, employment status and education level. Income was measured using equalised income variable deciles from lowest to highest. This measure considers different financial resource requirements of different households by adjusting total household income for the number of adults and children in the household. Employment was categorised using participants' reported activity in the week before the interview. Education was categorised using participants' reported highest qualification achieved at the time of the interview. People with missing employment or education data were excluded from the main analysis due to small numbers (n=45; 0.04% and n=105; 0.08%, respectively). A larger number of people had missing income information (n=19 007; 15.2%); therefore, a separate category of 'missing income' was included to investigate if they represent a different population.

### Statistical analysis
Descriptive statistics for people with and without reported opioid use was calculated. To analyse the effect of age on the likelihood of reported opioid use in groups of different SES, a semiparametric GAM was implemented in the following form:

$$\log\left(\frac{P(Y=1 \mid X=x)}{P(Y=0 \mid X=x)}\right) = \alpha + \beta X + f(age)(z_i)$$

In the parametric portion of the model, $\alpha$ is the intercept and $\beta$ is the vector of parameters associated with a set of explanatory variables $X$. The results are presented in ORs and the corresponding 95% CIs.

 Nowakowska M, *et al. BMJ Open* 2023;**13**:e057428. doi:10.1136/bmjopen-2021-057428

The non-parametric portion of the model is formed by the smoothing function $f$ for age which can vary for each category of SES measure $zi$. GAM is an extension of the commonly used Generalised Linear Model, which relaxes the hypothesis that the relationship between predictor and outcome variable is linear.[18] Instead, the relationship is modelled using smooth functions which can take any shape. Varying the shape of the smooth function by category of SES allows better interpretation of the relationship between age and opioid use in groups with different income, employment status and education levels. This relationship can be interpreted using the effective degrees of freedom (EDF) which can indicate whether the relationship is linear (EDF=1) or non-linear (EDF>1). Furthermore, graphical visualisation of the relationship is provided to aid the interpretation. Models were controlled for available demographics: sex and ethnicity.

As a sensitivity analysis, all analyses were performed separately for women and men to investigate if different patterns emerge. These analyses showed no noticeable difference and are presented in the online supplemental Tables SM1 and SM2, figures SM1–SM6. All analyses were performed using R V.3.6.3, and GAM estimations were performed using the *mgcv* package, with thin plate regression splines, for modelling the smooth term in age.[19] The number of basis dimensions $k$ was automatically selected through the *gam* () function and the adequacy of $k$ was assessed.

## Patient and public involvement

No patients were involved in the design, implementation or writing of this paper. The results will be disseminated to the appropriate audience.

## RESULTS

In total, 124 740 people met the inclusion criteria, with 2470 (1.98%) reporting opioid use across all waves (table 1). The mean age of opioid users was higher than non-users (59.00±15.38 vs 48.47±17.75). A higher proportion of women and white people were opioid users than men and people of any other ethnicity (2.23% vs 1.67% and 2.13% vs 0.86%, respectively). After adjusting for all covariates, in all models with the three SES indicators, women, white people and people living in urban areas were more likely to report opioid use (table 2).

Compared with people in the lowest income decile, people reporting income in the highest decile were significantly less likely to report opioid use (OR: 0.27, 95% CI: 0.21 to 0.36). Compared with people who were employed the week before the interview, unemployed people were not statistically different in the likelihood of reporting opioid use. However, as expected, those classified as ill or disabled were significantly more likely to report opioid use (OR: 16.50, 95% CI: 13.24 to 20.55). People performing domestic work were also more likely to report opioid use (OR: 1.60, 95% CI: 1.32 to 1.94). Those with higher education were less likely to report

**Table 1** Characteristics of all eligible participants from HSE 1997 to 2014

|  | Opioid non-users | Opioid users |
|---|---|---|
| Number (%) | 122 270 (98.02) | 2470 (1.98) |
| Age, mean (±SD) | 48.47 (±17.75) | 59.00 (±15.38) |
| **Gender** | | |
| Female, n (%) | 67 584 (97.77) | 1544 (2.23) |
| Male, n (%) | 54 686 (98.33) | 926 (1.67) |
| **Ethnicity** | | |
| White | 107 797 (97.87) | 2343 (2.13) |
| Non-white | 14 390 (99.14) | 125 (0.86) |
| **Income decile** | | |
| 1—lowest | 10 300 (97.41) | 274 (2.59) |
| 2 | 10 288 (97.30) | 286 (2.71) |
| 3 | 10 218 (96.63) | 356 (3.37) |
| 4 | 10 272 (97.15) | 301 (2.85) |
| 5 | 10 342 (98.34) | 231 (2.18) |
| 6 | 10 398 (98.74) | 175 (1.66) |
| 7 | 10 440 (98.74) | 133 (1.26) |
| 8 | 10 475 (99.07) | 98 (0.93) |
| 9 | 10 461 (98.94) | 112 (1.06) |
| 10—highest | 10 487 (99.19) | 86 (0.81) |
| Income missing | 18 589 (97.80) | 418 (2.20) |
| **Employment status** | | |
| Student | 4123 (99.66) | 14 (0.34) |
| Employed | 69 564 (99.33) | 465 (0.66) |
| Unemployed | 2367 (99.25) | 18 (0.75) |
| Ill or disabled | 4726 (86.87) | 714 (13.13) |
| Retired | 29 055 (96.46) | 1067 (3.54) |
| Domestic worker* | 11 321 (98.56) | 165 (1.44) |
| Other | 1071 (97.72) | 25 (2.28) |
| Employment missing | 43 (95.55) | 2 (4.44) |
| **Education†** | | |
| Higher education | 36 026 (98.79) | 441 (1.21) |
| A-levels | 13 896 (98.71) | 181 (1.29) |
| O-levels/GCSE | 31 358 (98.16) | 586 (1.83) |
| Foreign/other | 3624 (97.84) | 80 (2.16) |
| No qualifications | 31 238 (96.51) | 1130 (3.49) |
| Full time student | 6026 (99.19) | 49 (0.81) |
| Education missing | 102 (97.14) | 3 (2.86) |

*Any person engaged in domestic work within an employment relationship.
†The O-level and A-level examination certificates are the secondary and pre-university credentials in England, Wales and Northern Ireland. The O levels, or ordinary levels, typically represent a total of 11 years of study and mark the end of the secondary education cycle. A-levels, or advanced level qualifications, are subject-based qualifications (leading to university, further study, training or work), studied by students in Sixth Form, which refers to the last 2 years of secondary education (ages 16–18). A General Certificate of Secondary Education (GCSE) is a qualification normally taken by most UK students at the end of compulsory education.
HSE, Health Survey for England.

**Table 2** Semiparametric generalised additive models for the odds of reported prescription opioid use with smooth function for the effect of age, varying by SES measure

| Parametric coefficients | Opioid use (model including income) OR (95% CI) | Opioid use (model including employment status) OR (95% CI) | Opioid use (model including education) OR (95% CI) |
|---|---|---|---|
| Intercept | 0.009 (0.007 to 0.011)*** | 0.003 (0.002 to 0.003)*** | 0.01 (0.007 to 0.011)*** |
| Female (ref: male) | 1.32 (1.21 to 1.43)*** | 1.53 (1.40 to 1.67)*** | 1.33 (1.22 to 1.44)*** |
| White (ref: not white) | 2.24 (1.86 to 2.70)*** | 2.01 (1.67 to 2.42)*** | 1.94 (1.61 to 2.32)*** |
| Urban (ref: rural) | 1.27 (1.15 to 1.41)*** | 1.20 (1.08 to 1.33)*** | 1.29 (1.17 to 1.43)*** |
| Income decile (ref: 1— lowest) | | | |
| 2 | 0.93 (0.76 to 1.13) | | |
| 3 | 1.01 (0.82 to 1.24) | | |
| 4 | 0.80 (0.65 to 0.99)* | | |
| 5 | 0.55 (0.43 to 0.70)*** | | |
| 6 | 0.51 (0.40 to 0.64)*** | | |
| 7 | 0.38 (0.29 to 0.49)*** | | |
| 8 | 0.34 (0.26 to 0.44)*** | | |
| 9 | 0.40 (0.32 to 0.51)*** | | |
| 10—highest | 0.27 (0.21 to 0.36)*** | | |
| Income missing | 0.64 (0.53 to 0.77)*** | | |
| Economic status (ref: employed) | | | |
| Student | | 0.92 (0.36 to 2.35) | |
| Unemployed | | 1.61 (0.97 to 2.64) | |
| Ill or disabled | | 16.50 (13.24 to 20.55)*** | |
| Retired | | 1.60 (0.93 to 2.76) | |
| Domestic worker | | 1.60 (1.32 to 1.94)*** | |
| Other | | 3.14 (2.02 to 4.87)*** | |
| Education (ref: no qualifications) | | | |
| Higher education | | | 0.48 (0.41 to 0.57)*** |
| A-levels | | | 0.57 (0.47 to 0.70)*** |
| O-levels/GCSE | | | 0.71 (0.61 to 0.83)*** |
| Foreign/other | | | 0.63 (0.43 to 0.91)* |
| Student | | | 0.57 (0.44 to 0.82)** |
| **Non-parametric** | EDF | EDF | EDF |
| Income decile | | | |
| 1—lowest | 3.97*** | | |
| 2 | 6.08*** | | |
| 3 | 4.17*** | | |
| 4 | 3.31*** | | |
| 5 | 2.64*** | | |
| 6 | 2.51*** | | |
| 7 | 2.06*** | | |
| 8 | 1.52*** | | |
| 9 | 1.04*** | | |
| 10—highest | 1.01*** | | |
| Income missing | 3.30*** | | |
| Economic status | | | |

Continued

**Table 2** Continued

| Parametric coefficients | Opioid use (model including income) OR (95% CI) | Opioid use (model including employment status) OR (95% CI) | Opioid use (model including education) OR (95% CI) |
|---|---|---|---|
| Employed | | 1.01*** | |
| Student | | 1.002 | |
| Unemployed | | 1.002* | |
| Ill or disabled | | 3.12*** | |
| Retired | | 4.99** | |
| Domestic worker | | 2.74*** | |
| Other | | 1.76* | |
| Education | | | |
| Higher education | | | 2.03*** |
| A-levels | | | 2.93*** |
| O-levels/GCSE | | | 3.30*** |
| No qualifications | | | 4.30*** |
| Foreign/other | | | 1.01 ** |
| Student | | | 2.81 *** |

*P value <0.05, **p value <0.01, ***p value <0.001.
EDF, estimated degrees of freedom; GCSE, General Certificate of Secondary Education; SES, socioeconomic status.

opioid use than those with no qualifications (OR: 0.48, 95% CI: 0.41 to 0.57).

The effect of age on the likelihood of reporting opioid use was non-linear and significant in all income deciles as suggested by the statistically significant EDF. The higher the income decile, the relationship between age and probability of opioid use approached linear relationship with the EDF: 1.01, p<0.001 for highest income decile and EDF: 3.97, p<0.001 for lowest income decile. This suggests that in people with higher income, older people were more likely to report opioid use. However, in lower-income deciles, the relationship was non-linear. In particular, in people on the lowest income decile, the probability of opioid use increased with age until approximately 55 years old, after which it reduced (figure 1).

Concerning employment status, age had a statistically significant association with reported opioid use in people who were employed (EDF: 1.01, p<0.001), ill or disabled (EDF: 3.12, p<0.001), domestic workers (EDF: 2.74, p<0.001), and with lower significance levels, retired (EDF: 4.99, p<0.01), unemployed (EDF: 1.002, p<0.05) and other economic statuses (EDF: 1.76, p<0.05). In people who were employed and unemployed, the relationship approached linearity (EDF: 1.01, p<0.001 and EDF: 1.002, p<0.05, respectively), whereas in people who were ill or disabled, the distribution was slightly skewed towards people between 40 and 60 years old, and in domestic workers, the likelihood peaked at around 60 years of age (figure 2).

For all levels of education status, the effect of age on the odds of reporting opioid use was statistically significant. The effect was almost linear for people with foreign/

other qualifications (figure 3). In people with higher education, older age was associated with higher odds of reported opioid use. In contrast, in people with lower qualifications (A-levels, O-levels/General Certificate of Secondary Education) the odds of reported opioid use peaked around 60 years old slightly decrease after that. The odds of reported opioid use peaked earlier in people with no qualifications at around 50 years old.

## DISCUSSION

This study found that the relationship between age and the likelihood of reported opioid use changes with different measures of SES. In people with higher income and education, the relationship is more linear, increasing with age. In people on lower income with lower levels of education, the likelihood of reported opioid use peaks at around 40–60 years old and slowly decreases after that. Consistently with previous research, women, white people, those living in urban areas and those with lower levels of SES were more likely to report opioid use. The difference between employed and unemployed people was not statistically significant regarding the likelihood of reported opioid use. In both groups, the association with age was linear and increasing with age.

Existing studies found mixed results regarding the correlation between age and use of opioids; some studies found a higher prevalence in older populations[10] whereas others in the working-age population.[12] This study shows that SES may contribute to the non-linear relationship between age and opioid use. Shaw et al[20] suggest two pathways in which working conditions, strongly correlated

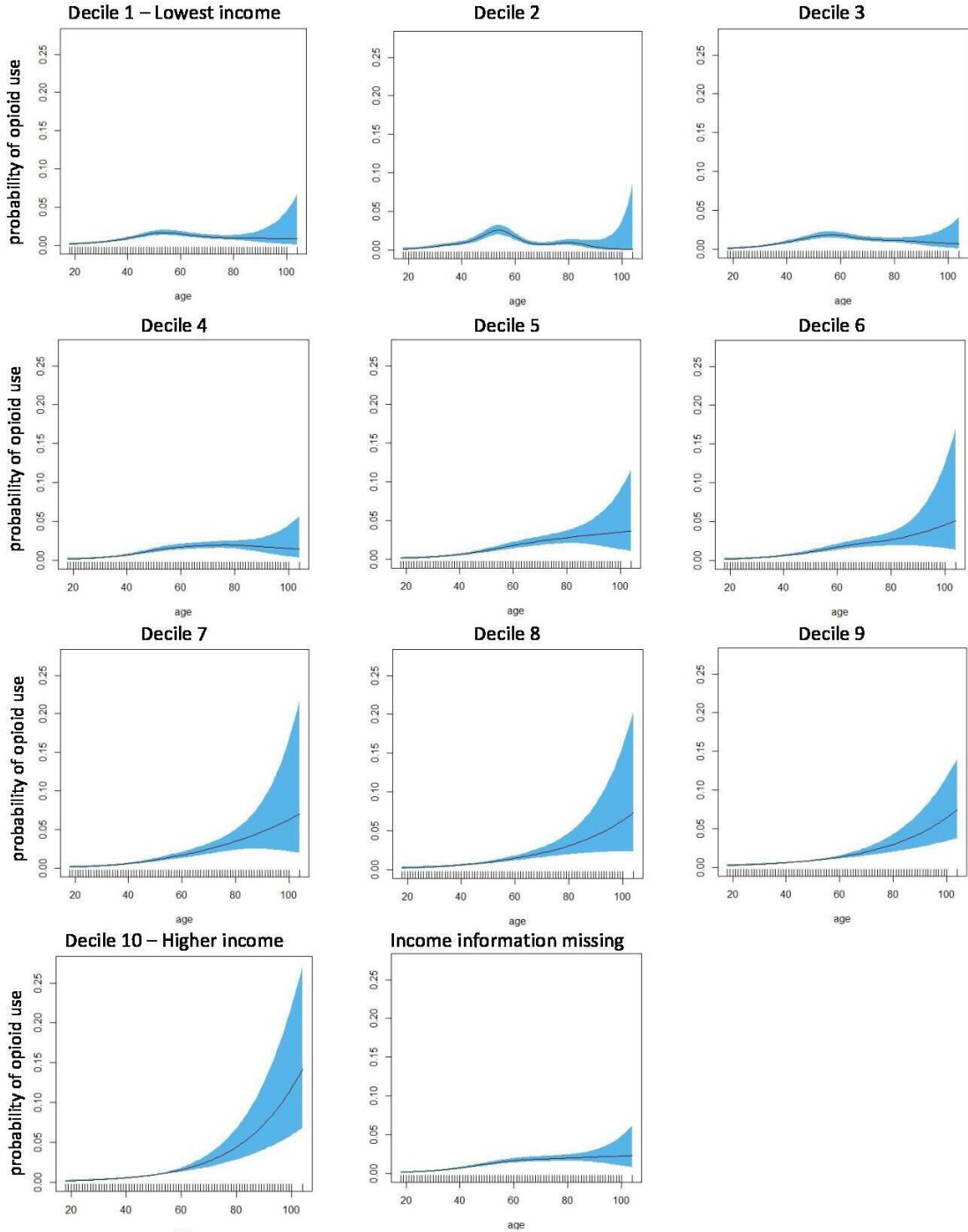

**Figure 1** Non-parametric estimation of the effect of age on likelihood of reported opioid use by income decile.

with income, related to opioid use. The first suggests that physical injury at work or cumulative trauma from strenuous labour leads to pain and the use of painkillers. The second relates to psychological stress from unmanageable job demands like time pressure or economic insecurity and job instability, leading to depression and anxiety, which may correlate with chronic opioid use. These adverse working conditions are expected in people with lower-paying jobs, explaining the observed higher prevalence of reported opioid use in working-age people with lower SES. Furthermore, people experiencing disability

or illness who are in a greater need of opioid treatment are more likely to be on lower income due to difficulties sustaining employment, regardless of age. Interestingly, we observe a reversal in the relationship between rurality and opioid use, compared with what has been reported in the USA.[21 22] However, this is not unexpected considering the different socioeconomic make-up in the rural USA and rural UK, with rural UK areas being generally much more affluent than urban conurbations, contrary to the situation in the USA.[23]

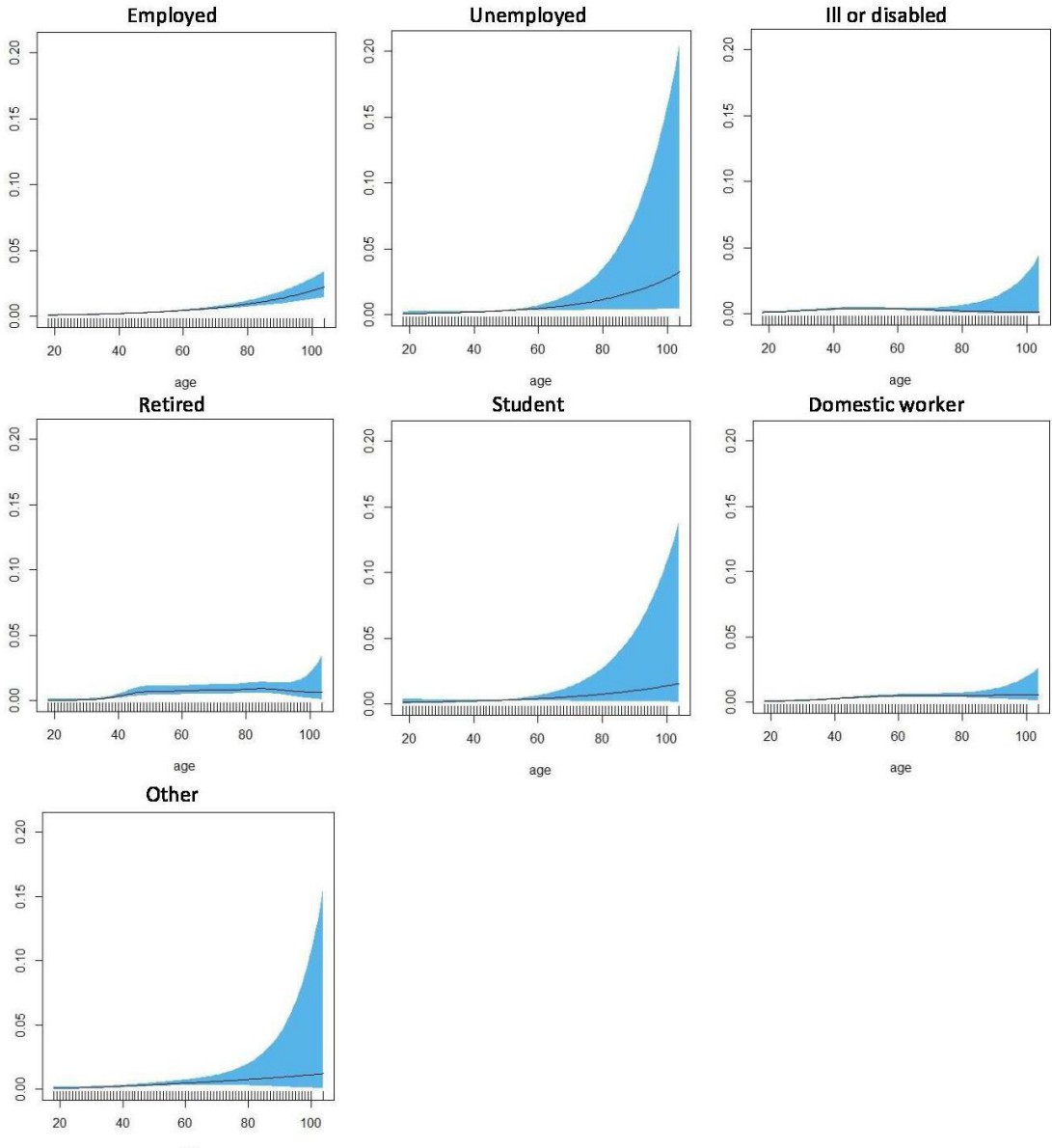

**Figure 2** Non-parametric estimation of the effect of age on likelihood of reported opioid use by economic status.

Our findings show an increase in reported opioid use with age in people with higher income and higher education. Reported opioid use in people with lower income, and also in people with no higher education, decreased after it peaked around 40–60 years of age. Two possible explanations have been identified in the literature. First, previous research found that the prevalence of musculoskeletal pain increases up to approximately 65 years old, after which it declines.[24] This has been linked to the decline of the adverse physical and mental effects of the workplace at the age of retirement and the lower prevalence of several diseases in surviving populations as people with chronic diseases tend to die early.[25] Furthermore, a review of pain management in the elderly showed that pain in the elderly might be untreated and misdiagnosed.[26] Conversely, a higher prevalence of opioid use would be expected in older people due to more complex health needs and the use of opioid analgesia in palliative care. Further research is needed to establish if older people from lower socioeconomic backgrounds are more likely to experience difficulties accessing adequate analgesia.

### Strengths and limitations

The statistical techniques used in this study allowed modelling the relationship between age and odds of reported opioid use in a non-linear manner and present results separately for people with different levels of SES. Various methods can examine the interaction between two variables, each with different strengths and limitations. A common approach includes an interaction term where the effect of a covariate of interest can vary depending on another variable.[27] Alternatively, multilevel models allow parameters of interest, including the intercept and

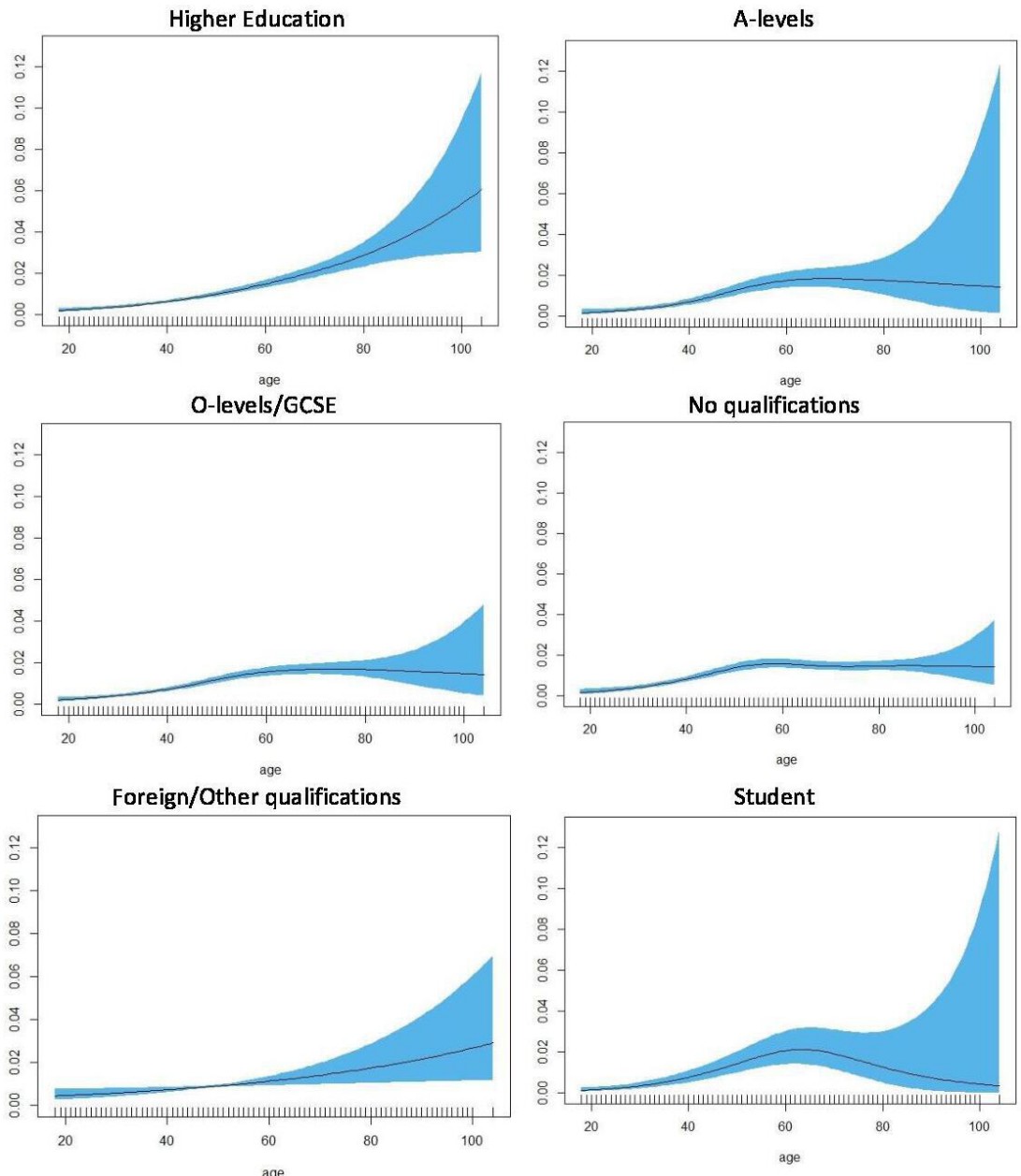

**Figure 3** Non-parametric estimation of the effect of age on likelihood of reported opioid use by education status

the slope of the regression line, to vary across more than one level.[28] However, these methods assume a linear relationship. Age can be modelled as categorical, but this can lead to complex models where multiple age categories interact with multiple SES categories, and information is difficult to interpret. Other techniques, including several machine learning methods, can produce highly accurate but difficult to interpret estimations. This study employed GAMs, a regression analysis method that allows exploration of the non-linearity of the relationship between age and opioid use within groups of different income, employment status and education levels. To our knowledge, this is the first study to employ this technique to explore the relationship between age and prescription opioid use in England.

However, some limitations should be acknowledged. First, the relationship between age and opioid use may be specific to different opioid strengths and durations. The data records in HSE did not capture the type of prescribed opioids; hence they did not adjust for the opioid type strength in the analysis. Second, opioid use was assessed based on self-reported data, which may be susceptible to recall and social desirability bias. However, this risk of bias was reduced by the involvement of qualified nurses in collecting medication data. Third, the low prevalence of opioid use meant that a limited number of covariates was included in the model. Some variables, such as the presence of chronic conditions, had to be excluded from the models, which could explain the different patterns of associations across socioeconomic strata. Fourth, the

HSE, although considered representative of the English population,[16] does not include some subgroups of the population, who are not living in private households, such as the homeless (who are at higher risk of opioid abuse) and those living in communal establishments (eg, residential and nursing homes). However, less than 2% of the population of England live in communal establishments,[29] while less than 0.5% are homeless.[30]

## Conclusion

Findings from this study suggest that, although the relationship between age and odds of opioid use in people with higher income and education approaches linearity, in people with lower SES, the probability of exposure peaks between 40 and 60 years old and decreases after that. This variability in the relationship between age and the probability of prescribed opioid use highlights different drivers in opioid prescribing across different income and educations strata. More research is needed into exploring patterns in opioid use in older people, particularly from disadvantaged socioeconomic backgrounds.

**Author affiliations**
[1]NIHR School for Primary Care Research, The University of Manchester, Manchester, UK
[2]Division of Informatics, Imaging and Data Sciences; Faculty of Biology, Medicine and Health, The University of Manchester, Manchester, UK
[3]NIHR Greater Manchester Patient Safety Translational Research Centre, The University of Manchester, Manchester, UK
[4]Division of Population Health, Health Services Research and Primary Care; Faculty of Biology, Medicine and Health, The University of Manchester, Manchester, UK
[5]Division of Pharmacy and Optometry; Faculty of Biology, Medicine and Health, The University of Manchester, Manchester, UK

**Contributors** All authors contributed to the design of the study. MN prepared and analysed the data. MN wrote the manuscript and MN, SSZ, L-CC, DA and EK all critically edited the manuscript. All authors approved the manuscript before submission. MN had full access to all the data in the study and had final responsibility for the decision to submit for publication.

**Funding** This study was funded as part of a PhD studentship from the National Institute for Health and Care Research (NIHR) School for Primary Care Research (SPCR) and NIHR Greater Manchester Patient Safety Translational Research Centre (award number: PSTRC-2016-003). The study represents independent research by the NIHR. The views expressed in this publication are those of the authors and not necessarily those of the NIHR or the Department of Health and Social Care. The study funders had no role in the study design, data collection, analysis or interpretation, in the writing of the paper or in the decision to submit the paper for publication.

**Competing interests** None declared.

**Patient and public involvement** Patients and/or the public were not involved in the design, or conduct, or reporting, or dissemination plans of this research.

**Patient consent for publication** Not applicable.

**Provenance and peer review** Not commissioned; externally peer reviewed.

**Data availability statement** Data are available in a public, open access repository. The data are freely available from NHS Digital: https://digital.nhs.uk/data-and-information/areas-of-interest/public-health/health-survey-for-england---health-social-care-and-lifestyles.

**ORCID iDs**
Magdalena Nowakowska http://orcid.org/0000-0003-1386-2534
Salwa S Zghebi http://orcid.org/0000-0002-7978-1094
Evangelos Kontopantelis http://orcid.org/0000-0001-6450-5815

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
