## [Reviewer comments · BMJ Open]

ARTICLE DETAILS

TITLE (PROVISIONAL)	Examining non-linearity in the association between age and reported Opioid Use, in different socioeconomic strata: Cohort Study Using Health Survey for England Waves from 1997 to 2014
AUTHORS	Nowakowska, Magdalena; Zghebi, Salwa; Chen, Li-chia; Kontopantelis, Evangelos

VERSION 1 – REVIEW

REVIEWER	Aytur, Semra Univ New Hampshire, Health Management and Policy
REVIEW RETURNED	03-Feb-202299

GENERAL COMMENTS	This paper adds to the literature by assessing whether the association between age and self-reported opioid use is nonlinear, and whether the association differs across SES levels. The strengths of the study include the large sample size and the longitudinal study design which utilizes the Health Survey for England waves 1997 to 2014 data. The use of a semi-parametric generalized additive model (GAM) is also a strength of the approach. Including data visualizations and sensitivity analyses are helpful components. However, I have the following concerns and questions about the study: 1) Can the authors please clarify whether the design is a cohort or a panel? From the description given, as a reader, this is difficult to assess. It seems that there were 'waves' of individuals sampled, but they were not necessarily the same individuals? Clarification from the authors on the design would be very helpful. Are there any population sub-groups that would likely be underestimated or not counted in the Health Survey for England, such as homeless individuals, who may also have high rates of opioid use?2) Although 'mediating effect' appears in the title, I do not see evidence of mediation analysis in the methods or the results provided. The analysis appears to be more about moderation than mediation. If the authors do wish to frame this work as mediation, this would need significant explanation and potentially additional analysis to present pathways of direct and indirect effects and the type of mediation model used. A simpler approach might be to amend the title to remove the words 'mediating effect'.3) The approach for addressing confounding variables is unclear. Can the authors please add more detail about which confounders or covariates were included, and what type of model building strategy was used? Is the list shown in Table 1 inclusive
--

	of all the variables in the model? The authors explain that the number of covariates had to be limited due to the low prevalence of self-reported opioid use. Please comment on whether other unmeasured confounders (in addition to comorbidities) could potentially impact the results. 4) It is interesting that the authors report (on p. 6) that persons living in urban areas were more likely to report opioid use. This contrasts with some studies conducted in other countries, such as the U.S., where high rates of opioid use are frequently observed in rural areas (and where the median age also tends to be higher in rural areas, along with greater socioeconomic disadvantage). Could the authors provide readers with more detail about how these patterns compare to those observed in other countries? 5) To provide greater context for readers, could the authors briefly describe the social and political context in England from 1997-2014? It would be helpful if the authors could explain whether there were any important policy changes during that time, for example those aimed at poverty alleviation, labor/wage changes, etc. Related to point #2, might any of these policy or environmental variables be considered important omitted variables or confounders? These issues may have important implications for discussing how the results vary by SES and age.
--	--

REVIEWER	Lam, Tina Monash University Eastern Clinical Research Unit, Monash Addiction Research Centre
REVIEW RETURNED	15-Feb-2022

GENERAL COMMENTS	I commend the authors on an interesting analysis of a highly relevant research question. The clarity of writing was superb. I have some minor comments below, but overall I think the paper was well considered and provides an elegant model for how SES modifies age-related shifts in pharmaceutical opioid use.  1. P4 L18 “provide crucially pain relief” - typo 2. Where word count allows, please use more specific terms such as “pharmaceutical opioid” or “prescription opioids” in the Introduction’s description of the background literature to exclude other types of opioids such as illicitly produced heroin or non-prescribed use of pharmaceutical opioids. 3. P5. It is noted that Health Survey for England (HSE) participants are randomly selected by household. It is unclear the proportion of potential participants that consent and participate. Can the authors please make a statement around the overall representativeness and scale of the HSE, especially as they might relate to over or under sampling certain sub-populations? 4. Can the authors confirm that individuals were only included in the cohort if they were able to contribute data to all 17 waves? If there was another minimum wave number, please state what that was and the proportion of participants who were able to contribute to all waves. 5. Table 2. Suggest adding a footnote or equivalent on the Education levels presented. As an international reader, I’m
---

	unfamiliar with the number of years of study “A levels” and “O levels/GSE” represent. 6. Similarly, I’m not sure what “Domestic worker” means – does this refer to a “homemaker” who has duties within their own home, or an individual who is employed to complete domestic duties in another person’s home? I’m assuming the former as the latter would just be classed as “Employed”, but it’d be useful to have a footnote with the definition the HSE used. 7. A mean and SD measure for age has been provided in Table 1. The x-axis on the models extends to 104 years. Could the authors please confirm the age distribution of the participants or provide raw numbers for the age groups (esp. as ~2% would be aged 90+)? 8. Fig 1. Please label the y-axis to likelihood or reported opioid use or similar (acknowledging some redundancy in the figure title). 9. P9 L26. “peaked around 60 years old slightly decrease after that” - typo 10. P10. L4. “Reported opioid use in people with lower income and no higher education decreased after it peaked around 40 to 60 years old”. Suggest clarifying this refers to two separate groupings as it made me wonder where the model was which combined Income and Education measures to define a subgroup of “people with lower income and no higher education”.
--	---

REVIEWER	Lacasse, Anaïs Université du Québec en Abitibi-Témiscamingue, Department of health sciences
REVIEW RETURNED	15-Feb-2022

GENERAL COMMENTS	The authors conducted an analysis of Health Survey for England (HSE) data to assess the association between age and opioid use when stratifying by various other socioeconomic variables. A semi-parametric generalized additive model was applied and showed that linearity of the association between age and opioid use varied across education level and income strata. Here are comments for authors to consider:  1. Abstracts – The conclusion should focus on the concrete implications of results. 2. Introduction – Although benefits of opioids for chronic noncancer pain remain modest, many chronic pain patients use such medications. Chronic pain is not addressed in the manuscript. 3. Introduction – Although the knowledge gap about the relationship between age and opioid use is underlined in the introduction, the relevance of conducting such a study should be better underlined. What is the clinical relevance concretely? Implications for researchers? 4. Methods, data source – Is the HSE a series of cross-sectional surveys or a longitudinal study conducted in the same participants? It should be clearly explained to the reader. 5. Methods, ethics – It is underlined that the HSE surveys are reviewed yearly by the Research Ethics Committee. Which one? Did the principal investigator’s institution Ethics Committee approve the project? Reference number? 6. Methods, study population – The authors underline that cancer patients are excluded and that complex sampling strategies are applied in the HSE (boost sample). More information should be provided about the representativeness of the study sample. Also, when surveys oversample some profiles of individuals, shouldn’t weights be used in the data analysis?
--

	7. Methods, variables and analysis – As underlined by the authors, lack of consideration for the type, the dose and the duration of opioid use is a major limitation. In addition, other potential confounding factors are not taken into account in the analysis (i.e., factors that could be associated with age and independently association with opioid use such as chronic pain and its characteristics, other medications used, etc.). 8. Methods, patient and public involvement statement – There must be a copy-paste error here. The authors refer to a systematic review and meta-analysis. Also the statement about results dissemination to the public is too vague. 9. Results – Prevalence of opioid use in Table 1 is 1.98%, I'm not sure to understand how it can be 2.23% across all waves? What explains this difference? 10. Results, Table 1 – When looking at age, mean values across users and non-users of opioids are presented. However, this is not the case for categorical variables. Following the logic of the presentation format for age, the proportion of women should be presented across users and non-users of opioids. But here it is the opposite, the prevalence of opioid use is presented according to the subgroups presented in the first column of the table. Presentation should be standardized across all Table 1's variables. 11. Results, Table 2 – How so the main independent variable (age) is not presented with its OR in the table? 12. Results, Table 2 footnotes – “Odds Ratio” is presented in its singular form vs. “Confidence Intervals” in its plural form. 13. Discussion – as underlined, the concrete implications of results should be discussed.
--	---

VERSION 1 – AUTHOR RESPONSE

Reviewer: 1

Dr. Semra Aytur, Univ New Hampshire

Comments to the Author:

This paper adds to the literature by assessing whether the association between age and self-reported opioid use is nonlinear, and whether the association differs across SES levels. The strengths of the study include the large sample size and the longitudinal study design which utilizes the Health Survey for England waves 1997 to 2014 data. The use of a semi-parametric generalized additive model (GAM) is also a strength of the approach. Including data visualizations and sensitivity analyses are helpful components.

Response: Thank you for your positive assessment.

However, I have the following concerns and questions about the study:

1) Can the authors please clarify whether the design is a cohort or a panel? From the description given, as a reader, this is difficult to assess. It seems that there were ‘waves’ of individuals sampled, but they were not necessarily the same individuals? Clarification from the authors on the design would be very helpful. Are there any population sub-groups that would likely be underestimated or not counted in the Health Survey for England, such as homeless individuals, who may also have high rates of opioid use?

Response: The title of the submitted paper is “The Nonlinear Association between Age and Reported Opioid Use and the Role of Socioeconomic Status: Cohort Study Using Health Survey for England Waves from 1997 to 2014”, where the study design is clearly described as per the STROBE statement.

The design is also mentioned in the abstract and the methods section. As with most cohorts of observational data, there are entries and exits during the study period.

Regarding the second point, the HSE is considered representative of the population of England, although some subgroups, not living in private households are necessarily excluded. We are now discussing this in the paper, having expanded the limitation section to state: “Fourth, the HES, although considered representative of the English population,[15] does not include some subgroups of the population, who are not living in private households, such as the homeless (who are at higher risk of opioid abuse) and those living in communal establishments (e.g. residential and nursing homes). However, less than 2% of the population of England live in communal establishments,[28] while less than 0.5% are homeless.[29]”

2) Although ‘mediating effect’ appears in the title, I do not see evidence of mediation analysis in the methods or the results provided. The analysis appears to be more about moderation than mediation. If the authors do wish to frame this work as mediation, this would need significant explanation and potentially additional analysis to present pathways of direct and indirect effects and the type of mediation model used. A simpler approach might be to amend the title to remove the words ‘mediating effect’.

Response: Thank you, that is a valid point and we have removed the single mention of mediation.

3) The approach for addressing confounding variables is unclear. Can the authors please add more detail about which confounders or covariates were included, and what type of model building strategy was used? Is the list shown in Table 1 inclusive of all the variables in the model? The authors explain that the number of covariates had to be limited due to the low prevalence of self-reported opioid use. Please comment on whether other unmeasured confounders (in addition to comorbidities) could potentially impact the results.

Response: Thank you, for raising this point. The model included the variables listed in table 2 and we have made this clearer in the methods section. We only controlled for available demographics (sex and ethnicity) since these are available covariates that are strongly correlated with age (sex) and deprivation (ethnicity). In the context of exploring varying patterns of the association between age and opioid prescribing across different socioeconomic strata, we feel that our analyses are robust and confounding is less of an issue. Yes, the characteristics of the people in the more deprived areas will be driving the differences in the patterns, but this is not confounding, and we would not want to control for anything that will explain this difference in the first instance. There are, clearly, different relationships between age and prescribing, by deprivation, and we cannot envisage a confounding mechanism that would argue that this is not the case – so there is no “impact” on the results and our conclusions. However, we do acknowledge that the data are limited and we stop short of attempting to explain, through the data, why that is the case. The comorbidities, if possible, would be used as a second analysis, to attempt to explain the different patterns. The IMD is an aggregate of 37 socio-economic indices, and any of these could have a larger role that is underplayed through the use of the aggregate, in addition to the health of the person (using the presence of chronic conditions as a proxy of health).

4) It is interesting that the authors report (on p. 6) that persons living in urban areas were more likely to report opioid use. This contrasts with some studies conducted in other countries, such as the U.S., where high rates of opioid use are frequently observed in rural areas (and where the median age also tends to be higher in rural areas, along with greater socioeconomic disadvantage). Could the authors provide readers with more detail about how these patterns compare to those observed in other countries?

Response: Thank you, we have now expanded the discussion to acknowledge this: “Interestingly, we observe a reversal in the relationship between rurality and opioid use, compared to what has been reported in the USA. [20,21] However, this is not unexpected considering the different socioeconomic make-up in the rural USA and rural UK, with rural UK areas being generally much more affluent than urban conurbations, contrary to the situation in the USA.[22]”

5) To provide greater context for readers, could the authors briefly describe the social and political context in England from 1997-2014? It would be helpful if the authors could explain whether there were any important policy changes during that time, for example those aimed at poverty alleviation, labor/wage changes, etc. Related to point #2, might any of these policy or environmental variables be considered important omitted variables or confounders? These issues may have important implications for discussing how the results vary by SES and age.

Response: Thank you for the suggestion, but to describe the social and political context of England over 18 years is a major challenge and beyond the scope of the paper. There are too many policy interventions, global and national shocks, and different governments, to try to associate with the observed trends and associations in a 4000 words paper. Regarding confounding, please see our response to point 3. We do hypothesise on the mechanisms that may explain this, in the third paragraph of the discussion.

Reviewer: 2

Dr. Tina Lam, Monash University Eastern Clinical Research Unit Comments to the Author:

I commend the authors on an interesting analysis of a highly relevant research question. The clarity of writing was superb. I have some minor comments below, but overall I think the paper was well considered and provides an elegant model for how SES modifies age-related shifts in pharmaceutical opioid use.

Response: Thank you for your positive assessment.

1) P4 L18 “provide crucially pain relief” - typo

Response: Thank you, corrected.

2) Where word count allows, please use more specific terms such as “pharmaceutical opioid” or “prescription opioids” in the Introduction’s description of the background literature to exclude other types of opioids such as illicitly produced heroin or non-prescribed use of pharmaceutical opioids.

Response: Thank you, we have made changes to clarify this.

3) P5. It is noted that Health Survey for England (HSE) participants are randomly selected by household. It is unclear the proportion of potential participants that consent and participate. Can the authors please make a statement around the overall representativeness and scale of the HSE, especially as they might relate to over or under sampling certain sub-populations?

Response: Thank you, this related to a similar point raised by reviewer #1. Please see our response to point 1b of reviewer #1, and the resulting additions to the paper.

4) Can the authors confirm that individuals were only included in the cohort if they were able to contribute data to all 17 waves? If there was another minimum wave number, please state what that was and the proportion of participants who were able to contribute to all waves.

Response: No that was not the case, the households are randomly selected in every wave from all eligible. Thus, although technically the same person may be selected the probability is very small. We do provide a key reference to the HSE early in the methods section where more information is available. We have rephrased the relevant description to clarify this point to: “Participants in the HSE are randomly chosen, for each wave, from all private households’ addresses in England”.

5) Table 2. Suggest adding a footnote or equivalent on the Education levels presented. As an international reader, I’m unfamiliar with the number of years of study “A levels” and “O levels/GSE” represent.

Response: Thank you for highlighting this oversight on your part. We have added the following clarification as a footnote for Table 1:

“The O- and A-Levels examination certificates are the secondary and pre-university credentials in England, Wales and Northern Ireland. The O Levels, or Ordinary Levels, typically represent a total of 11 years of study and mark the end of the secondary education cycle. A-levels, or Advanced level qualifications, are subject-based qualifications (leading to university, further study, training, or work), studied by students in Sixth Form, which refers to the last two years of secondary education (ages 16–18). A General Certificate of Secondary Education (GCSE) is a qualification normally taken by most UK students at the end of compulsory education.”

6) *Similarly, I’m not sure what “Domestic worker” means – does this refer to a “homemaker” who has duties within their own home, or an individual who is employed to complete domestic duties in another person’s home? I’m assuming the former as the latter would just be classed as “Employed”, but it’d be useful to have a footnote with the definition the HSE used.*

Response: Thank you. Homemakers are captured in the “Unemployed” category. Domestic worker is any person engaged in domestic work within an employment relationship. We have clarified that in a footnote in Table 1.

7) *A mean and SD measure for age has been provided in Table 1. The x-axis on the models extends to 104 years. Could the authors please confirm the age distribution of the participants or provide raw numbers for the age groups (esp. as ~2% would be aged 90+)?*

Response: Thank you. Homemakers are captured in the

8) *Fig 1. Please label the y-axis to likelihood or reported opioid use or similar (acknowledging some redundancy in the figure title).*

Response: Thank you, we have added a y-axis label. We are happy to liaise with the production team to finalise the figures.

9) *P9 L26. “peaked around 60 years old slightly decrease after that” – typo*

Response: Thank you, we have now changed to: “peaked at around 60 years of age”.

10) *P10. L4. “Reported opioid use in people with lower income and no higher education decreased after it peaked around 40 to 60 years old”. Suggest clarifying this refers to two separate groupings as it made me wonder where the model was which combined Income and Education measures to define a subgroup of “people with lower income and no higher education”.*

Response: Thank you, we have now changed to “Reported opioid use in people with lower income, and also in people with no higher education, decreased after it peaked around 40 to 60 years of age”

Reviewer: 3

Dr. Anaïs Lacasse, Université du Québec en Abitibi-Témiscamingue Comments to the Author:

The authors conducted an analysis of Health Survey for England (HSE) data to assess the association between age and opioid use when stratifying by various other socioeconomic variables. A semi-parametric generalized additive model was applied and showed that linearity of the association between age and opioid use varied across education level and income strata.

Here are comments for authors to consider:

1) *Abstracts – The conclusion should focus on the concrete implications of results.*

Response: Thank you, the conclusions section now reads as “The relationship between age and the probability of prescribed opioid use varies greatly across different income and education strata, highlighting different drivers in opioid prescribing across population groups. More research is needed into exploring patterns in opioid use in older people, particularly from disadvantaged socioeconomic backgrounds.”

2) *Introduction – Although benefits of opioids for chronic noncancer pain remain modest, many chronic pain patients use such medications. Chronic pain is not addressed in the manuscript.*

Response: Thank you, that is a valid point. We have expanded the introduction to address this. We now say: “Opioid painkillers are effective analgesics that can provide crucially pain relief for patients with acute or cancer pain, which are often still used for the treatment of chronic non-cancer pain, although evidence for the latter indicate higher risk of short-term harms and limited or no benefits compared to nonopioid therapy.[1]”

3) *Introduction – Although the knowledge gap about the relationship between age and opioid use is underlined in the introduction, the relevance of conducting such a study should be better underlined. What is the clinical relevance concretely? Implications for researchers?*

Response: Thank you, we have now expanded the introduction to clarify the aims of the paper and the potential implications: “Different associations between age and opioid prescribing, across socio-economic strata, would potentially highlight the presence of different health needs and drivers for prescribing. This would have implications for research, since we would need to understand the underlyingly cause or causes of this variation, to inform policy and improve patient care.”

4) *Methods, data source – Is the HSE a series of cross-sectional surveys or a longitudinal study conducted in the same participants? It should be clearly explained to the reader.*

Response: Technically, it is a cohort (longitudinally) study but it includes a new random sample of patients (with replacement, but chances that someone will contribute twice are very small). We have made this clearer in the study population section, in response to a similar point raised by reviewer #1: “Participants in the HSE are randomly chosen, for each wave, from all private households’ addresses in England”.

5) *Methods, ethics – It is underlined that the HSE surveys are reviewed yearly by the Research Ethics Committee. Which one? Did the principal investigator’s institution Ethics Committee approve the project? Reference number?*

Response: Thank you, we have now specified the ethics committee in the paper, East Midlands - Nottingham 2, please see: <https://www.hra.nhs.uk/planning-and-improving-research/application-summaries/research-summaries/the-health-survey-for-england-2016-to-2019/>. The data is freely available, after anonymisation, and no further approvals are required. Ethical approval is needed only if stored blood or serum samples from HSE participants are to be analysed. We have expanded the relevant section (data source) to clarify that no further approvals were required.

6) *Methods, study population – The authors underline that cancer patients are excluded and that complex sampling strategies are applied in the HSE (boost sample). More information should be provided about the representativeness of the study sample. Also, when surveys oversample some profiles of individuals, shouldn’t weights be used in the data analysis?*

Response: Thank you, we do provide a key reference which describes the cohort in full: Mindell J, Biddulph JP, Hirani V, *et al.* Cohort profile: The health survey for england. *Int J Epidemiol* 2012;**41**:1585–93. doi:10.1093/ije/dyr199. Regarding the representativeness, this is a valid point and we have expanded the limitations section to discuss this further, as per the request of reviewer #1. Finally, there two common approaches to analysing surveys. In the first (as in the HSE), certain populations are oversampled to more closely match the target population – in this case analysis is as standard, usually unweighted regression. In the second, weights may be provided by the survey for the analyses, to more closely match the target population – in this case analysis is usually done with weighted regression techniques.

7) *Methods, variables and analysis – As underlined by the authors, lack of consideration for the type, the dose and the duration of opioid use is a major limitation. In addition, other potential confounding factors are not taken into account in the analysis (i.e., factors that could be associated with age and independently association with opioid use such as chronic pain and its characteristics, other medications used, etc.).*

Response: Thank you, this relates to a similar point raised by reviewer #1. Please see our response to point 3 by reviewer #1. In short, we would not want to control for anything that will explain this difference in the first instance. There are, clearly, different relationships between age and prescribing, by deprivation, and we cannot envisage a confounding mechanism that would argue that this is not the case – so we can argue, robustly, that there are different patterns in the association between age and prescribing probability, across socio-economic strata on the results and our conclusions. However, we do acknowledge that the data are limited and we stop short of attempting to explain, through the data, why that is the case. We have also expanded the limitations section to explain this issue further.

8) *Methods, patient and public involvement statement – There must be a copy-paste error here. The authors refer to a systematic review and meta-analysis. Also the statement about results dissemination to the public is too vague.*

Response: Thank you for your careful reading. Apologies, indeed, this was from the previous paper of the PhD, and it has now been corrected.

9) *Results – Prevalence of opioid use in Table 1 is 1.98%, I'm not sure to understand how it can it be 2.23% across all waves? What explains this difference?*

Response: Thank you, this was a mistake. We were reporting the female rather than the overall prevalence in the text, and that has now been corrected to be reported consistently as 1.98%.

10) *Results, Table 1 – When looking at age, mean values across users and non-users of opioids are presented. However, this is not the case for categorical variables. Following the logic of the presentation format for age, the proportion of women should be presented across users and non-users of opioids. But here it is the opposite, the prevalence of opioid use is presented according to the subgroups presented in the first column of the table. Presentation should be standardized across all Table 1's variables.*

Response: Thank you, if we understand this point correctly, we think the reviewer is arguing for percentages to be reported across columns, allowing a comparison of the categorical characteristics across opioid users and non-users. This is a slightly different approach, that would report the distribution of each covariate within the groups – we are aiming to report the percentage of users within each category. We would prefer to keep the table as is, but this is not due to inertia and we report the table the reviewer requested below. We are happy to follow the editor's recommendation when it comes to the final version of the table

Table 1: Characteristics of all eligible participants from HSE 1997-2014

	Opioid non-users	Opioid users
Number (%)	122,270	2,470
Age, mean (\pm SD)	48.47 (\pm 17.75)	59.00 (\pm 15.38)
Gender		
Female, N (%)	67,584 (55.27%)	1,544 (62.51%)
Male, N (%)	54,686 (44.73%)	926 (37.49%)
Ethnicity		
White	107,797 (88.22%)	2,343 (94.94%)
Non white	14,390 (11.78%)	125 (5.06%)

Income decile		
1 – lowest	10,300 (8.42%)	274 (11.09%)
2	10,288 (8.41%)	286 (11.58%)
3	10,218 (8.36%)	356 (14.41%)
4	10,272 (8.40%)	301 (12.19%)
5	10,342 (8.46%)	231 (9.35%)
6	10,398 (8.50%)	175 (7.09%)
7	10,440 (8.54%)	133 (5.38%)
8	10,475 (8.57%)	98 (3.97%)
9	10,461 (8.56%)	112 (4.53%)
10 - Highest	10,487 (8.58%)	86 (3.48%)
Income missing	18,589 (15.20%)	418 (16.92%)
Employment Status		
Student	4,123 (3.37%)	14 (0.57%)
Employed	69,564 (56.89%)	465 (18.83%)
Unemployed	2,367 (1.94%)	18 (0.73%)
Ill or disabled	4,726 (3.87%)	714 (28.91%)
Retired	29,055 (23.76%)	1,067 (43.20%)
Domestic worker†	11,321 (9.26%)	165 (6.68%)
Other	1,071 (0.88%)	25 (1.01%)
Missing Employment	43 (0.04%)	2 (0.08%)
Education*		
Higher Education	36,026 (29.46%)	441 (17.85%)
A-levels	13,896 (11.37%)	181 (7.33%)
O-levels/GCSE	31,358 (25.65%)	586 (23.72%)
Foreign/other	3,624 (2.96%)	80 (3.24%)
No qualifications	31,238 (25.55%)	1,130 (45.75%)
Full time student	6,026 (4.93%)	49 (1.98%)
Education Missing	102 (0.08%)	3 (0.12%)

11) Results, Table 2 – How so the main independent variable (age) is not presented with its OR in the table?

Response: This is due to the nature of the model, which attempts to model age in a non-parametric way and hence age is not one of the explanatory variables usually denoted as X in a regression model. More information on this is available in the methods section.

12) Results, Table 2 footnotes – “Odds Ratio” is presented in its singular form vs. “Confidence Intervals” in its plural form.

Response: We feel this is appropriate since when reporting a point estimate we would say, “the Odds ratio was X and its confidence intervals were A to B”

13) Discussion – as underlined, the concrete implications of results should be discussed.

Thank you, the conclusions section now reads as “Findings from this study suggest that, although the relationship between age and odds of opioid use in people with higher income and education

approaches linearity, in people with lower SES, the probability of exposure peaks between 40 to 60 years old and decreases after that. This variability in the relationship between age and the probability of prescribed opioid use, highlights different drivers in opioid prescribing across different income and education strata. More research is needed into exploring patterns in opioid use in older people, particularly from disadvantaged socioeconomic backgrounds.”

VERSION 2 – REVIEW

REVIEWER	Bai, Ray University of South Carolina
REVIEW RETURNED	28-Nov-2022

GENERAL COMMENTS	The study is interesting and uses statistically sound methodology. The conclusions and inferences drawn from the study are appropriate. I just have a few minor comments and questions: 1) What was the smooth class used for modeling the smooth term in age? The mgcv package has many different choices (e.g. thin plate regression splines, B-splines, cubic regression splines, etc.). Was the "default" thin plate regression splines used for this analysis? The authors should indicate this in the manuscript. 2) Analysis with smoothing terms can be somewhat sensitive to the choice of k, or the number of basis functions used to estimate the smooth function. For example, you can specify the basis dimension k in the gam() function as, e.g. <code>mod <- gam(y ~ s(x1, k=5) + x2 + x3, data = data)</code>. Did the authors let the gam() function choose k for them, or did the authors specify this themselves? And did the authors check if the basis dimension k used was appropriate using the <code>gam.check()</code> function to ensure that the basis dimension used is adequately large? If <code>gam.check()</code> determines that k is not adequate, then the authors should increase k, i.e. they should manually set the argument k to be sufficiently large and possibly conduct some sensitivity analyses to different choices of k. Inclusion of these details could be helpful to ensure the soundness of the methodology.
--

VERSION 2 – AUTHOR RESPONSE

Reviewer: 4

Dr. Ray Bai, University of South Carolina Comments to the Author:

The study is interesting and uses statistically sound methodology. The conclusions and inferences drawn from the study are appropriate. I just have a few minor comments and questions:

1) What was the smooth class used for modeling the smooth term in age? The mgcv package has many different choices (e.g. thin plate regression splines, B-splines, cubic regression splines, etc.). Was the "default" thin plate regression splines used for this analysis? The authors should indicate this in the manuscript.

Response: Thank you for raising this. We did indeed use the “default” thin plate regression splines and have now added that information to the manuscript.

2) Analysis with smoothing terms can be somewhat sensitive to the choice of k , or the number of basis functions used to estimate the smooth function. For example, you can specify the basis dimension k in the `gam()` function as, e.g. `mod <- gam(y ~ s(x1, k=5) + x2 + x3, data = data)`. Did the authors let the `gam()` function choose k for them, or did the authors specify this themselves? And did the authors check if the basis dimension k used was appropriate using the `gam.check()` function to ensure that the basis dimension used is adequately large? If `gam.check()` determines that k is not adequate, then the authors should increase k , i.e. they should manually set the argument k to be sufficiently large and possibly conduct some sensitivity analyses to different choices of k .

Response: Thank you, the `gam()` function was used and k was adequate. We have added that to the paper as well.

VERSION 3 – REVIEW

REVIEWER	Bai, Ray University of South Carolina
REVIEW RETURNED	20-Dec-2022
GENERAL COMMENTS	Thank you for addressing my questions.